# Antibiotic-Related Changes in Microbiome: The Hidden Villain behind Colorectal Carcinoma Immunotherapy Failure

**DOI:** 10.3390/ijms22041754

**Published:** 2021-02-10

**Authors:** Tsvetelina Velikova, Boris Krastev, Stefan Lozenov, Radostina Gencheva, Monika Peshevska-Sekulovska, Georgi Nikolaev, Milena Peruhova

**Affiliations:** 1Department of Clinical Immunology, University Hospital Lozenetz, Sofia University St. Kliment Ohridski, Kozyak 1 Str., 1407 Sofia, Bulgaria; 2Clinic of Medical Oncology, MHAT Hospital for Women Health Nadezhda, 1330 Sofia, Bulgaria; bmk83@abv.bg (B.K.); radostina_gencheva@mail.bg (R.G.); 3Laboratory for Control and Monitoring of the Antibiotic Resistance, National Centre for Infectious and Parasitic Diseases, 26 Yanko Sakazov Blvd, 1504 Sofia, Bulgaria; dr_lozenov@abv.bg; 4Department of Gastroenterology, University Hospital Lozenetz, Sofia University St. Kliment Ohridski, Kozyak 1 Str., 1407 Sofia, Bulgaria; mpesevska93@yahoo.com (M.P.-S.); mmp@mail.bg (M.P.); 5Faculty of Biology, Sofia University St. Kliment Ohridski, 1407 Sofia, Bulgaria; g_nikolaev@abv.bg

**Keywords:** checkpoint inhibitors, antibiotics, microbiome, colorectal carcinoma, immunotherapy, gut microbiota, cancer

## Abstract

The interplay between drugs and microbiota is critical for successful treatment. An accumulating amount of evidence has identified the significant impact of intestinal microbiota composition on cancer treatment response, particularly immunotherapy. The possible molecular pathways of the interaction between immune checkpoint inhibitors (ICIs) and the microbiome can be used to reverse immunotherapy tolerance in cancer by using various kinds of interventions on the intestinal bacteria. This paper aimed to review the data available on how the antibiotic-related changes in human microbiota during colorectal cancer (CRC) treatment can affect and determine ICI treatment outcomes. We also covered the data that support the potential intimate mechanisms of both local and systemic immune responses induced by changes in the intestinal microbiota. However, further better-powered studies are needed to thoroughly assess the clinical significance of antibiotic-induced alteration of the gut microbiota and its impact on CRC treatment by direct observations of patients receiving antibiotic treatment.

## 1. Introduction

The interplay between drugs and microbiota is critical for successful treatment. Therefore, pharmacomicrobiomics was proposed as a modern approach for assessing the interaction between medicines and microbiota composition. A growing amount of evidence suggests that intestinal microbiota composition significantly impacts cancer treatment response, particularly immunotherapy. It has been shown that specific intestinal bacteria may promote or suppress immune checkpoint inhibitor (ICI) effectiveness. The possible molecular pathways of such an interaction can be used to reverse immunotherapy tolerance in cancer by using various kinds of interventions on intestinal bacteria [1]. 

However, the intestinal microbiome can alter bioavailability, activity, and toxicity by transforming medication [2]. This widely-accepted concept lays stress on the importance of the balance between beneficial and harmful bacteria rather than the presence or absence of individual bacterial species. Numerous factors have been identified as altering temporarily or permanently the structure of intestinal microbiota, the key ones being diet, antibiotics, other pharmaceuticals, immune system factors, and cancer [3].

Changes in the typical intestinal microbiota composition have been linked—but not confined—to inflammation, a well-known cancer trait. The microbiome–immune system–cancer axis needs further evaluation, especially regarding microbiota’s influence on different cancer therapies, including immunotherapy. Targeted immune-mediated treatments have been a crucial point in recent developments regarding solid tumor treatment. ICIs have been one of the most significant groups of treatments to impact the whole field of oncology. ICIs have demonstrated clinical benefits in various cancers, including colorectal cancer (CRC) with microsatellite instability (dMMR/MSI-H) in the metastatic setting [4,5]. Therefore, the increasing role of ICIs in treating cancer has led to further analyses of the factors predicting their response—mostly, which factors can lead to failure of ICI treatment.

Direct and indirect alterations of the microbiome composition and metabolic capacity as a consequence of antibiotic usage have been examined and documented thoroughly [6]. However, changes in the gut microbiota caused by antibiotics may also alter the number and functions of immune cells in the intestines, leading to systemic inflammatory responses in the body [7]. In line with this, it is not surprising that the effectiveness of immunotherapy may vary in response to alterations in microbiota composition [8,9].

This paper aimed to review the data available on how the antibiotic-related changes in human microbiota during CRC treatment can affect and determine ICI treatment outcomes. We also cover the data that support the potential intimate mechanisms of both local and systemic immune responses induced by changes in the intestinal microbiota.

## 2. Microbiome, Local and Systemic Immune Responses, and Cancer

The intensive crosstalk between the host gut microbiota and the intestinal immune system is essential for the maturation of immune cells. The human microbiome is also involved in education and enabling the immune system to recognize potentially hazardous bacteria, thus avoiding invasion and infection. 

The effect of the gut microbiome on the response to immune checkpoint blockade of different cancer types has been shown [10]. In regard to CRC, it is considered that the aggressiveness and treatment response are determined by various immune cells inside the tumor and in the tumor microenvironment. It has been established that lymphoid and myeloid cells, fibroblasts, and endothelial cells are involved. The critical participation of cytotoxic CD8+ T cells, dendritic cells, tumor-associated macrophages, and cancer-associated fibroblasts play a central role as modulators and even drivers of tumor heterogeneity [11]. The immune cell populations within the tumor and the tumor environment are not the only players that encourage tumor development. It has been documented that gut microbiome dysbiosis with loss of protective bacterial populations and the enrichment of microbial communities can encourage cancer. Therefore, it is essential to understand the dynamic changes in CRC patients’ gut microbiome to clarify the whole process of CRC carcinogenesis.

Recent studies have shown that patients with CRC have an altered gut microbiome compared to healthy controls. For example, an overabundance of several intestinal bacterial organisms, including *Fusobacterium nucleatum*, enterotoxigenic *Bacteroides fragilis* (ETBF), and *Peptostreptococcus anaerobius*, have been reported to promote tumor proliferation in CRC carcinogenesis [12]. Furthermore, the correlation between the bacteria mentioned above and inflammation and the tumor shield for immune attack has been suggested.

The critical effect of commensal microbes on cancer patients’ prognosis has also been reported in recent studies. Mima et al. revealed the association between the abundance of *Fusobacterium nucleatum* DNA in CRC tissue and a shorter survival in a large patient cohort study. Moreover, the amount of *F. nucleatum* DNA may potentially serve as a prognostic biomarker in clinical outcomes [13]. Moreover, Yu et al. revealed the role of *F. nucleatum* in fostering chemoresistance in patients with CRC by inducing autophagy, which fails treatment or provokes disease recurrence [14]. Recently, we gathered data on *F. nucleatum* and its connection with microbiota dysbiosis, the progression of CRC, the transformation of conventional adenoma to CRC, and the serrated carcinoma pathway [15].

One of the prominent causes that increases the risk of accelerated proliferation and deleterious mutations is the degradation of mucin by pathogens, resulting in biofilm formation on the epithelial surface and adverse immune responses [16]. Thus, one would speculate that biofilm invasion deep into colonic crypts would accelerate carcinogenesis. In patients with biofilms, the risk of developing CRC is higher compared to those without biofilms. These bacterial biofilms are also observed in normal colon mucosa, but have been associated with decreased colonic epithelial cell E-cadherin expression and increased activation of IL-6 and STAT3 in epithelial cells, as well as increased proliferation of crypt epithelial cells [16]. We also recently established IL-6 as a crucial cytokine of equal importance for both inflammation and tumor development, suggesting that IL-6 is a significant tumor promoter during the early stages of CRC [17].

Although the exact mechanisms remain to be discovered, a growing number of preclinical animal model studies and clinical trials of immunotherapies suggest that the host microbiome is a critical determinant for the variable host responses to different therapy modalities [18]. Yu et al. demonstrated that the same type of pro-inflammatory cells induced due to dysbiosis leads to increased T cell exhaustion in the tumor microenvironment [19]. Gut microbiota dysbiosis also increases the intestinal barrier’s permeability, favoring bacterial translocation, macrophages activation, and the consequent establishment of chronic pro-tumorigenic inflammation. 

Studies have already identified specific commensal flora members that exert significant microbiome-dependent control on anti-tumor immunity, including immune system priming and the response to ICIs. However, it is difficult to identify the single most significant member of the flora for tumor favoring. Still, several species contribute to cancer through different mechanisms, and some species have detrimental effects on anti-tumor treatment. It has been demonstrated that normal gut microbiota might enhance the anti-tumor activity of ICIs by promoting the secretion of IL-12 by local dendritic cells and by changing the local repertoires of Th1 cells to express the intestinal chemokine receptors CCR9 and CXCR3. It has been reported that the anti-tumor potential of CD8+ cells is also affected by the intestinal microbiota, where anti-tumor mechanisms depend on increased IFN-γ production and CD8+ count within the tumor [9,20]. For example, *Bifidobacterium* has been shown to improve anti-tumor immunity in vivo, both alone and in combination with anti-PD-L1 immunotherapy through modulation of dendritic cell function and the subsequently improved effector function of tumor-specific CD8+ T cells [21].

Furthermore, lysates of *Lactobacillus acidophilus* also have enhanced anti-tumor efficacy of anti-CTLA-4 (cytotoxic T lymphocyte-associated protein 4) blocking antibody in mice CRC models, associated with increased CD8+ T cells, decreased numbers of T regulatory cells (Tregs), and decreased M2 macrophages in the tumor microenvironment [22]. Interestingly, bacterial genotoxins from *Bacteroides fragilis*, *Campylobacter jejuni*, and *Fusobacterium nucleatum* could promote CRC development in patients via activation of CD4+ Th17 cell responses, mTOR signaling, and the NF-kB pathway, respectively [23,24]. Additionally, bacteria such as ETBF and *Fusobacterium nucleatum* can impair the effectiveness of traditional chemotherapy and ICIs. The variations in treatment results between right-sided and left-sided colon cancer can be explained by biofilms in the right colon [16], as discussed above.

## 3. Novel Insights on the Antibiotic-Induced Changes in the Microbiome

The human intestinal flora has been subject to vigorous studies in recent years, mainly due to the advances in genetic techniques allowing for sequencing studies of unculturable bacteria. Metagenomic analyses of gut microbiome changes induced by antibiotics still involve mostly small (<100) cohorts of patients/volunteers, allowing much room for intragroup variability. These limitations were the primary and intrinsic weaknesses of the above analyses. Moreover, metagenomic analyses from animal models of the human intestinal microbiota are not a suitable replacement; for some more recent classes of antibiotics, such as carbapenems and polymixins, human metagenomic data are scarce, if any. Further work on larger populations will be needed to affirm the trends.

However, advanced technologies have allowed thorough examination of the gut microbiota’s genera level, demonstrating the domination of the taxa Bacteroides and Firmicutes, compared to relatively lower shares of Proteobacteria, Actinobacteria, and Fusobacteria [25].

Thus, antibiotics exert various neglected side effects on gut microbiota, affecting immune cell development, function, and regulation. When antibiotics diminish the beneficial gut microbiota, changes in the numbers and function of naïve cells, Th1/Th2 cells, Th17, and T regulatory cells occur [26]. Some of these alterations exert systemic effects in the organism, such as increased susceptibility to infections and sepsis. The systemic immune dysfunction found in antibiotic-treated patients is associated with impaired defense against pathogens, dysregulated toll-like receptor (TLR) signaling, reduced expression of antimicrobial peptides, low IgA production in the mucosa, decreased expression of IFN-γ (causing impaired clearance of viruses), etc. [27]. Except for the main drugs altering the microbiome, i.e., antibiotics, several other common non-antibiotic medicines have been related to disrupted gut microbiota composition and function.

In line with this, the microbiome changes during antibiotic therapy that affect the immune system may influence the immunotherapy efficacy, including ICIs for CRC. The issue is complicated because one has to analyze how different antibiotics impact the intestinal flora and how they interact with the immune mechanisms, especially in cancer and cancer treatment.

Claesson et al. focused on general antibiotic usage (regardless of the class of antibiotics). They discovered a trend of shifting the balance toward Bacteroides at the expense of Firmicutes, Actinobacteria, and Proteobacteria [28]. The same study analyzed the use of a broad wide-spectrum macrolide antibiotic (clarithromycin) explicitly in a short time frame of three months. They found that this specific antibiotic shifted the balance toward Firmicutes at a lower dose of 250 mg and away from major phyla at a higher dose of 500 mg. Furthermore, the study illustrated that the balance between favorable and unfavorable bacteria in terms of anti-tumor immune modulation is variably disturbed by different antibiotic classes, and even by their doses [28].

Another small study assessed the effect of a second-generation cephalosporin (cefprozil) on healthy volunteers’ intestinal flora, which is also a widely used antibiotic in outpatient settings. The findings of the study demonstrated high individual variability of the response to the treatment. The trends showed a reduction of Bacteroides representatives, and an increase of *E. coacae* and *Lachnoclostridium bolteae* [29]. The study also observed incomplete recovery to the initial state in several of the exposed subjects. Thus, lower microbiome diversity and the prevalence of Bacteroides enterotype may remain long after the antibiotic treatment.

A deeper look into the available data for alterations of the human microbiota under antibiotic treatment shows that the composition changes mostly follow the pattern of intrinsic resistance of the respective families to the individual antibiotics. Although there is evidence of the beneficial and detrimental members of the intestinal flora in view of the ICI therapy, data regarding species that are not pathogenic are limited in terms of antibiotic susceptibilities and are particularly problematic for unculturable or hardly culturable species such as *Akkermansia muciniphila*. 

The relationship between medicines and the microbiome is further extended by allowing some medications for CRC to be metabolized by microbiota into toxic metabolites or inactive molecules. For example, the drug availability and efficacy of monoclonal antibodies against PD-1 (programmed cell death protein 1) and its ligands PD-L1 and CTLA-4 may depend on gut microbiota composition [10,30].

A healthy microbiome has anti-tumor activities that can also impair the therapeutic success of the treatment, including cancer therapy with ICIs. Cancer immunotherapy can be enhanced or suppressed by the gut microbiome’s overall effects on the host immune system [31]. Furthermore, Routy et al. demonstrated that the use of antibiotics before, during, and after PD-L1 or PD-1 inhibition is linked to poorer prognosis and decreased progression-free survival [32]. Other anti-cancer drugs, such as daunorubicin and 5-fluorouracil, are associated with antimicrobial activity themselves [33].

Thus, standard chemotherapy success is often based on intact immune responses. These data support the hypothesis that intestinal microbiota can also modulate the therapy types. Therefore, cancer treatment success might depend on immune and microbiome interactions.

A schematic picture of ICI interactions with microbiota and immune cells is presented in Figure 1.

## 4. Immune Checkpoint Inhibitors in Regard to Microbiome and Antibiotics

Immune checkpoint inhibitors (ICIs) are a new class of systemic anti-tumor agents, rendering the patient’s immune system to attack tumor cells. There are basically three types of ICIs currently implemented in clinical practice: anti-PD-1, anti-PD-L1, and anti-CTLA-4 monoclonal antibodies [34]. These relatively novel drugs have led to significant progress in managing metastatic tumors such as malignant melanoma, non-small cell lung cancer (NSCLC), and renal cell cancer [35].

The interactions of ICIs with the host microbiome are emphasized when *Bacteroides fragilis*, *Bifidobacterium breve*, and *Bifidobacterium longum* contribute to ICI actions in CRC. The synergic effects include an increase of tumor-infiltrating lymphocytes and enhancement of tumor dendritic cells [20,21].

*Akkermansia muciniphila* and *Faecalibacterium pausnitzii* have demonstrated significant benefits in recruiting CCR9+CXCR3+CD4+ T lymphocytes to the tumor microenvironment through the mediation of IL-12 [32,36,37]. This is even more intriguing in the CRC context, considering that the effects mentioned above are most significant locally in the intestinal mucosa and the regional mesenteric lymphatic basin. *Enterococcus hirae* has been demonstrated to increase the CD8+/Treg ratio in tumors, with potential benefits for anti-tumor activity in immunotherapy [8]. Direct clinical studies have demonstrated a correlation between response to ICIs in cancer patients and the intestinal microbiota structure, assessed through metagenomic shotgun analysis [37].

Apart from beneficial interactions with individual intestinal microbiota members, there are also such interactions related to inhibition of ICI anti-tumor activity and respective treatment failure, such as *Fusobacterium nucleatum*, *Roseburia intestinalis*, *Prevotella* spp., *Proteobacteria* spp., and *Clostridales* spp. It has been shown that their mechanisms of action include inhibition of TLR4 on NK cells, reduced intratumoral CD8+ T cell recruitment, and reduced IFN-γ mediation [25].

On a clinical level, the existing data also support the hypothesis that gut microbiota might be relevant to the effectiveness of ICI cancer treatment. Moreover, drugs that are known modifiers of the intestinal flora, such as antibiotics, could impact ICI therapy by changing the gut microbiome, as we discussed earlier. 

For the time being, reports assessing the “microbiome–antibiotics–ICI” axis come mainly from retrospective studies or as a secondary endpoint or ad hoc analysis of clinical trials in melanoma, non-small cell lung cancer, and kidney or urothelial carcinoma. Due to the comparatively limited number of clinical trials assessing ICIs in CRC, so far, no report has addressed the issue specifically in these patients. Therefore, any speculation on microbiome–ICI interactions can only be extrapolated from the available data on the clinical evidence of the relevance of the microbiome and antibiotics to immunotherapy efficacy in other tumors.

Working on the preclinical evidence that certain bacterial strains in the gut correlate with ICI anti-tumor activity [21], one of the first to associate microbiota to the clinical outcomes of checkpoint inhibitors in humans was Matson et al. [36]. In a small cohort of 42 advanced-stage melanoma patients, ten different bacterial species were identified between anti-PD-1 therapy responders and non-responders. Furthermore, the authors verified their clinical observations by in vivo mice models, showing that gut colonization with responder-specific bacteria elicits tumor response to PD-1 inhibition. Their results also support preclinical data pointing at *Bifidobacterium* as a genus associated with ICI’s clinical benefit [36].

Almost at the same time, Routy et al. [32] reported a metagenomic analysis of feces from NSCLC (*n* = 60) and renal cell cancer patients (*n* = 40) who were to commence ICI therapy. The investigators managed to correlate clinical outcomes not only with specific bacterial strains, but also with the overall abundance and diversity of the gut microbiome. In their paper, the authors presented evidence that higher levels of bacterial diversity in stool samples correspond to favorable clinical outcomes in terms of six-month progression-free survival (PFS). When looking at distinct species in the feces, the most significant correlation with clinical benefit was shown by *Akkermansia muciniphila*. This was validated in additional cohorts of lung (*n* = 27) and renal cancer (*n* = 26) patients. Moreover, a higher bacterial number before ICI therapy initiation is associated with better tumor response and a PFS longer than three months.

Again, but this time in melanoma patients (*n* = 112), Gopalakrishnan et al. [10] demonstrated that diversity in the gut microbiome correlates significantly with objective response (complete, partial, or stable disease lasting for at least six months) to ICI treatment, as assessed by Response Evaluation Criteria in Solid Tumors (RECIST). Bacterial diversity in stool samples also correlated with PFS. Higher diversity was observed in those patients achieving longer PFS. As to the composition differences between microbiomes from responders vs. non-responders, the authors reported the Ruminococcaceae family and the *Faecalibacterium* genus as the most prevalent among responders, in contrast to species from the Bacteroidales order, which were abundant in the stools of ICI non-responders [10]. This and other studies (Chaput et al., 2017 [38]; Fukuoka et al., 2018 [39]; Maia et al., 2018 [40]) reporting an association between ICI efficacy and gut microbiome composition are presented in Table 1.

On the other side, as a drug class that strongly influences the gut’s microbial composition, antibiotics are righteously accused of interfering with ICI efficacy. More convincing and unambiguous data of their impact on the clinical outcome of checkpoint inhibitors may be obtained from direct observations on patients receiving antibiotic treatment during ICI therapy. Unfortunately, such reports are scarce, only retrospective, and unable to discriminate between the effects of different classes of antibiotics. However, one of the most extensive studies addressing the problem is an analysis of 303 patients with metastatic melanoma (*n* = 201), NSCLC (*n* = 56), and renal cell cancer (*n* = 46) [41]. Nearly one-third of them (*n* = 94) received antibiotics before or during ICI therapy. The most used antibiotics were beta-lactams and macrolides. The results of a multivariate analysis demonstrated significantly shorter PFS and overall survival (OS) in those patients who received antibiotic treatment as opposed to those who did not: PFS 97 days vs. 178 days, respectively, and OS 317 days vs. 651 days, respectively. The report also suggested that patients treated with antibiotics before initiating ICI treatment had shorter PFS and OS than those treated after ICI initiation [41]. The overall conclusion was that the altered gut microbiome, as a result of antibiotic treatment, was responsible for the more inferior results of immunotherapy, regardless of cancer type and patient’s clinical condition. 

In the paper by Routy et al. [32], which we cited earlier in this review paper, the authors included 249 patients with NSCLC (*n* = 140), RCC (*n* = 67), and urothelial carcinoma (*n* = 42), all treated with ICIs. Sixty-nine (28%) received antibiotic treatment (beta-lactam inhibitors, fluoroquinolones, or macrolides) within a time interval of two months before to one month after ICI initiation. The PFS and OS were significantly shorter in the antibiotic-treated group. This was independent of tumor type or other adverse clinicopathological features.

One of the most comprehensive studies assessing the interaction between antibiotics and ICIs so far is a meta-analysis performed by Huang et al. [42]. They summarized data from 19 studies and more than 2700 patients. The authors revealed a significant association between antibiotic usage and more unfavorable PFS and OS. Moreover, they found this relationship valid across different tumor types and independent of antibiotic administration time.

To the best of our knowledge, Khan et al. [43] performed the only analysis on a relatively large patient cohort, featuring gastrointestinal carcinomas (16 of 242) too, including a total of 242 patients assessed by antibiotic usage before or after initiation of ICIs. This study confirmed lower overall response rate in patients who received antibiotic treatment within 60 days after starting checkpoint inhibitors. 

Further better-powered studies discriminating between different antibiotics will be needed to fully understand the real-world clinical significance of antibiotic alteration of the gut microbiota and its impact on ICI treatment in CRC and other tumor types. We also have to admit that ICI effectiveness could depend on microbiome changes different from those observed in the gut only. The tumor microenvironment (TME) is another theater where bacterial flora confronts anti-tumor immunity [44]. This could be particularly relevant in gastrointestinal tumors, where the microbiome in the tumor microenvironment may not necessarily resemble that in the gut lumen. To date, data about the real impact of TME microbiome on ICI treatment clinical outcomes are lacking.

## 5. Conclusions

Apart from viruses, which are long known for their oncogenic potential, it turns out that bacteria are also involved in cancerogenesis, but in a much more complex manner: by modulating local and systemic anti-tumor immune responses. This is especially relevant for cancers of the gastrointestinal tract, where this issue is not related to a single “culprit,” but mostly to the fine balance in a community of microorganisms, properly termed by some as our “second organism.” Owing to its effect on a variety of disorders, including cancer, the microbiome attracts tremendous interest. Its function is increasingly evident when related to cancer therapies, especially when the intestinal microbiome regulation can impact the effectiveness and adverse effects of different types of cancer treatments. 

Technological progress has enabled us to dissect the gut microbiome and get into the very intricate compositional changes induced by different antibiotics. Going further down the road, we already know how this reflects on inflammation and adaptive immunity, so important for the anti-tumor effect of both chemotherapy and immunotherapy.

Despite global recommendations against random antibiotic usage, their wise and rational application still remains a challenge, especially nowadays, with the world fighting against newly emerged infectious threats. In this review, we discussed the consequences of antibiotic treatment, which, although not so directly harmful as resistance itself, could be responsible for adverse outcomes in patients with malignancies, including colorectal cancer. In contrast to the constitutional predictors of response, such as MSI, if the microbiome proves its significance for ICI efficacy, it could give us not only another biomarker, but a whole new tool to improve the clinical benefit of immunotherapy. For the time being, the research on gut bacteria, cancer immunity, and treatment provides a clue to their interaction, but to be able to implement it in practice, microbiome assessment and manipulation should be adopted in the design of future randomized trials. 

The factors and treatments that affect the gut microbiome themselves are incredibly crucial. Convincing and direct data for the impact of antibiotic treatment, specifically on the clinical outcomes of ICI treatments, are scarce and have been unable to discriminate between the effects of different classes of antibiotics so far. Further better-powered studies are needed to thoroughly assess the clinical significance of antibiotic alteration of the gut microbiota and its impact on CRC treatment by direct observations on patients receiving antibiotic treatment.

## Figures and Tables

**Figure 1 ijms-22-01754-f001:**
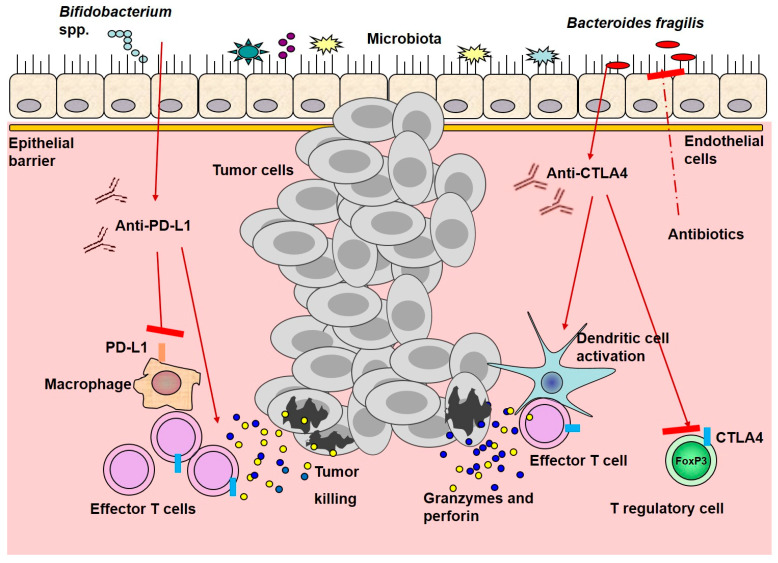
Immune checkpoint inhibitor interactions with microbiota and immune cells. Anti-PD-L1 treatment cooperates with resident *Bifidobacterium* spp., leading to activation of dendritic cells, promoting the activation, expansion, and function of T effector cells. Anti-CTLA4 promotes the enrichment of resident *Bacteroides* spp. And enhances dendritic and effector T cell activation, while suppresses T regulatory cells function. All of these mechanisms improve anti-tumor efficacy, in contrast to antibiotics that might decrease it. Red arrows represent stimulation, whereas red arrows with red rectangle—inhibition.

**Table 1 ijms-22-01754-t001:** Association of ICIs and the gut microbiome in humans.

			Bacterial Enrichment	
Study	Tumor	ICI	Responders	Non-Responders
Chaput et al., 2017 [38]	Malignant melanoma	Ipilimumab	*Faeca**libacterium*; other *Firmicutes*	*Bacteroidetes*
Matson et al., 2018 [36]	Malignant melanoma	Nivolumab	*Enterococcus faecium*; *Collinsella aerofaciens*; *Bifidobacterium adolescentis*; *Klebsiella pneumoniae*; *Veillonella parvula*; *Parabacteroides merdae*; *Lactobacillus* sp.; *Bifidobacterium longum*	*Ruminococcus obeum;* *Roseburia intestinalis*
Routy et al., 2018 [32]	NSCLC; RCC; urothelial cancer	Anti-PD-1; anti-PD-L1	*Akkermansia muciniphila*; *Enterococcus hirae*	NR
Frankel et al., 2017 [37]	Malignant melanoma	Ipilimumab + nivolumab; pembrolizumab	*Bacteroides caccae; Faecalibacterium prausnitzii; Bacteroides thetaiotamicron*; *Holdemania filiformis; Dorea formicogenerans*	NR
Gopalakrishnan et al., 2018 [10]	Malignant melanoma	Anti-PD-1	*Ruminococcaceae*, *Faecalibacterium*	*Bacteroides thetaiotaomicron*, *Escherichia coli*, and *Anaerotruncus colihominis*
Fu Fukuoka et al., 2018 [39]	NSCLC; Gastric cancer	Anti-PD-1	*Ruminococcaceae*	NR
Maia et al., 2018 [40]	RCC	Nivolumab	*Roseburia* spp. and *Faecalibacterium* spp.	NR

ICI, immune checkpoint inhibitor; NSCLC, non-small cell lung cancer; NR, not reported; RCC, renal cell cancer; PD-1, programmed cell death protein 1.

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
