# Peer review of "Antibiotic-Related Changes in Microbiome: The Hidden Villain behind Colorectal Carcinoma Immunotherapy Failure"

_ijms, 2021, doi:10.3390/ijms22041754_

Round 1

Reviewer 1 Report

The paper is extremely difficult to read. Despite proper vocabulary, there is no ‘story’. Facts are mixed and unstructured. Authors need to think about some structure.. As now the statements about ICI are intermixed with references to tumour initiation, sentences about signalling pathways and then back to treatment effects.

Text is very wordy and contains a lot of sentences without message. The overall ‘density’ of the information is also low.

Author Response

  • Thank you for your time to evaluate our paper. We would have been grateful if the critic was more constructive and detailed. We tried our best to revise our manuscript and make it more “readable” and “story-telling.” Besides, we rearranged some text passages to provide clarity and logical flow and increase our paper's scientific quality.

Reviewer 2 Report

This manuscript " Antibiotic-related changes in microbiome: the hidden villain 
behind colorectal carcinoma immunotherapy failure" may be improved if the author addresses the following comments.

Major criticisms:

The part between 116-129 need to be rewrite, it's not clear!

The chapter conclusions should also be extended with additional informations. The conclusion need to be clear. Perhaps 3 short conclusion can be added.

Minor:

The English written style should be revised by a English native speaker and the language requires some reconsideration in order to remove grammar and spelling inaccuracies and to make the manuscript more formal.

Please recheck the References order.

Thanks for opportunity of reading the article.

Author Response

This manuscript " Antibiotic-related changes in microbiome: the hidden villain behind colorectal carcinoma immunotherapy failure" may be improved if the author addresses the following comments.

  • Thank you for your feedback and the good overall evaluation of our paper.

Major criticisms:

The part between 116-129 need to be rewrite, it's not clear!

  • Thank you for the comment. We have rewritten the passage to make it clear, more precise, and consistent.

The chapter conclusions should also be extended with additional informations. The conclusion need to be clear. Perhaps 3 short conclusion can be added.

  • We agree that the conclusion section would benefit if three concluding statements have been included. We have revised the conclusion, taking into account your recommendations.

Minor:

The English written style should be revised by a English native speaker and the language requires some reconsideration in order to remove grammar and spelling inaccuracies and to make the manuscript more formal.

  • Our colleague Georgi Nikolaev, who is also an author in the paper, as well as a Certified Translator /English-Bulgarian-English/, heavily revised the text to improve the style, clarity, and natural flow of the language.

Please recheck the References order.

  • Thank you for noticing this issue. We have reordered and edited the reference list.

Thanks for opportunity of reading the article.

  • Thank you for the valuable comments that helped us to improve the paper`s quality.

Round 2

Reviewer 1 Report

The text demonstrates significant improvement, the overall flow in much better and easier to follow.

However, giving the amount of the information, I still think that the overall approach, used by authors (i.e very 'wordy' narrative) is not appropriate nor well implemented. This comment, however, rather reflects my personal impression and shell not be considered as criticism to the scientific content of the review.